# Synthesis of Hydroxyapatite (HAp)-Zirconia Nanocomposite Powder and Evaluation of Its Biocompatibility: An In Vitro Study

**Vignesh Raj Sivaperumal** [1]**, Rajkumar Mani** [2]**, Veerababu Polisetti** [3],*⬤**, Kanakaraj Aruchamy** [4],*⬤ **and Taehwan Oh** [4],*

1   Department of Biomedical Engineering, PSG College of Technology, Coimbatore 641004, India
2   Department of Physics, PSG College of Arts and Science, Coimbatore 641014, India
3   Wallenberg Wood Science Center, Department of Fibre and Polymer Technology, School of Engineering Sciences in Chemistry, Biotechnology and Health, KTH Royal Institute of Technology, SE-100 44 Stockholm, Sweden
4   School of Chemical Engineering, Yeungnam University, Gyeongsan 38541, Korea
*   Correspondence: vpo2@kth.se (V.P.); a.kanakaraj@yu.ac.kr (K.A.); taehwanoh@ynu.ac.kr (T.O.)

**Abstract:** A potential material for dental restorations and bone replacements is calcium phosphate (CaP)-based ceramic material. Nevertheless, its limited ability to withstand thermal processing and weak mechanical strength prevents it from being used in hard tissue engineering. Hydroxyapatite has been extensively used as a CaP-based biomaterial in prosthetic applications. On the other hand, zirconia is an inorganic material that combines outstanding mechanical capabilities with bioinert characteristics. In the present investigation, we demonstrated the reinforcement of zirconia in biomimetic hydroxyapatite (HAp) using a specially designed stir-type hydrothermal reactor to improve the biocompatibility and mechanical stability of bare hydroxyapatite. X-ray diffraction (XRD) analysis showed distinct peak shifts around $31°$ and $60°$, which confirmed the formation of a nanocrystalline HAp-Zirconia composite without any intermediate phases. The size of the synthesized nanocomposite was found to be 30 nm using TEM. Further, the d-spacing value calculated from high-resolution transmission electron microscope (HRTEM) images corresponded to the distinct planes of the HAp (211) and zirconia (311) phases, respectively, in the composite powder. The in vitro cytotoxicity study revealed excellent biocompatibility with MG-63 human osteoblasts. Hence, the zirconia reinforced hydroxyapatite (HZ1) prepared in the present work could be utilized as a successful approach in a variety of hard tissue engineering applications.

**Keywords:** hydroxyapatite; zirconia; nanocomposite; hydrothermal; biocompatibility

## 1. Introduction

### 1.1. The Significance of Biocompatible Materials in Biomedical Applications

Significant progress has been achieved in medical research and technology over the last few decades. Medical devices have been made and improved using a variety of materials in an effort to enhance their capabilities and increase patients' safety [1]. Substantial efforts have been put into developing biocompatible materials for biomedical applications [2–5]. Artificial biomaterials are being used more often to repair or replace broken bones. An ideal biomaterial for repairing or replacing injured bone tissue should have mechanical properties that are osteoinductive, osteoconductive, resorbable, and tunable to those of the bone environment. The use of calcium phosphate (CaP)-based biodegradable materials in reconstructive surgery of the bone has received a lot of attention to date [6].

### 1.2. Hydroxyapatite

Hydroxyapatite has been widely used as a CaP-based biomaterial in prosthetic applications owing to its excellent bioactivity, biocompatibility, corrosion resistance, chemical

stability, and lack of in vivo immune system interference or rejection [7–9]. Recent developments in nanoscience have rekindled studies into nanoscale HAp synthesis in order to precisely describe the small-scale features of HAp. Due to its excellent biocompatibility and capacity for bone integration, it has been claimed that nano-HAp would be an optimal biomaterial. The development of HAp biomedical materials has profited greatly from advances in nanotechnology [8]. When HAp is being implanted in vivo, the existence of a bone-like layer, such as apatite, on the surface of HAp is essential because it may create a bone-bond between the implant and the live bone tissue [9,10]. According to reports, the apatite surface on the bone could act as a good platform for osteoblastic development, proliferation, and differentiation. When combined with live cells, this surface helps to generate new bone [10]. Since it is the primary component of both human and animal bones and teeth, and can also stimulate bone development, it has emerged as the standard in medical treatment [11,12]. Both collagen and biological apatite, two elements that combine to form bone, are organic components.

### 1.3. Composites of Hydroxyapatite

For superior mechanical qualities, structural engineering of synthetic HAp is thus required, coupled with compositional matching. It has therefore been an intriguing area of study in recent years to add second-phase ceramic elements to the HAp matrix to produce products with increased strength and toughness [13]. Therefore, various materials such as alumina [14], titania [15], and silica [16] nanoparticles have been incorporated in order to increase the mechanical efficiency of HAp in hard tissue engineering. The effect of these nanoparticles in the formation of HAp-derived composites shows promising biomechanical properties for orthopedic applications [17].

### 1.4. The Hydroxyapatite-Zironia Composite

In particular, zirconia is an inorganic substance with excellent mechanical qualities. Due to its exceptional biocompatibility, robustness, and promotion of oral rehabilitation with good aesthetic and biological qualities, it has drawn considerable interest as a novel ceramic material for dental implants [18]. It also has bio-integrative components that are close to the ceramic inherent to bone, which expedites the improvement of the surface of bone minerals [19]. The addition of low-weight-percent bioactive zirconia refines the microstructure of HAp [20], reduces the crystallization temperature of HAp [21], enhances bioresorbability [22], and encourages osteoblast cell attachment [23] and apatite formation both in vitro and in vivo [24,25].

### 1.5. Methods of Preparation

Therefore, several synthesis methods, such as milling [26], precipitation methods [27], microwave irradiation [22], and the RF suspension plasma spray process [28], have been used to form homogeneous HAp-Zirconia composite powders. Among them, the hydrothermal method provides the necessary physico-chemical properties in the final product. The benefits of this method include faster interactions between fluid and solid species, lower operating temperatures, improved nucleation control, an increased reaction rate, increased dispersion, improved shape control, no pollution (because the process is carried out in an isolated system), improved reaction kinetics, and the production of phase pure and homogeneous materials [29].

### 1.6. Drawbacks of the Previous Approaches

However, during the utilization of the hydrothermal method, generally, a two-step process is required to prepare any composite material. The first step is to prepare one of the individual components of the composite and then, in the second step, while preparing another component of the composite, the component prepared in the first step will be combined with it to prepare the final composite. The disadvantage is that it is both time- and energy-consuming.

### 1.7. Objectives and Advantages of the Present Work

The current approach can overcome this problem because it consists of a single step. Due to the specific design of the hydrothermal reactor, the second step is eliminated and the precursor solution of the second component can be introduced while the instrument is in progress. Hence, the objective of the present work was to prepare a HAp-Zirconia composite by utilizing a specially designed hydrothermal reactor. This type of study is scarce in the literature. A HAp-Zirconia nanocomposite was prepared under optimized conditions using this uniquely crafted hydrothermal instrument and systematically studied for its physico-chemical, thermal, and mechanical properties. The microstructure and thermomechanical properties of the prepared nanocomposites were characterized using X-ray diffraction (XRD), transmission electron microscopy (TEM), thermogravimetric analysis (TGA), differential scanning calorimetric (DSC) analysis, elemental mapping techniques, and a nanoindentation study. In addition, we also evaluated the cytotoxicity of the synthesized HZ1 against MG-63 osteosarcoma cell lines.

## 2. Materials and Methods

The starting material for the preparation of the HAp-Zirconia composite included calcium nitrate tetrahydrate ($Ca(NO_3)_2 \cdot 4H_2O$), diammonium hydrogen phosphate (($NH_4)_2HPO_4$), zirconyl nitrate ($ZrO(NO_3)_2 \cdot H_2O$), and a 25% ammonia solution. All the chemical reactions were carried out in double-distilled water.

### 2.1. Synthesis of the HAp-Zirconia Composite

The process of preparing the HAp-Zirconia composite in hydrothermal conditions under autogenous pressure is shown in Scheme 1.

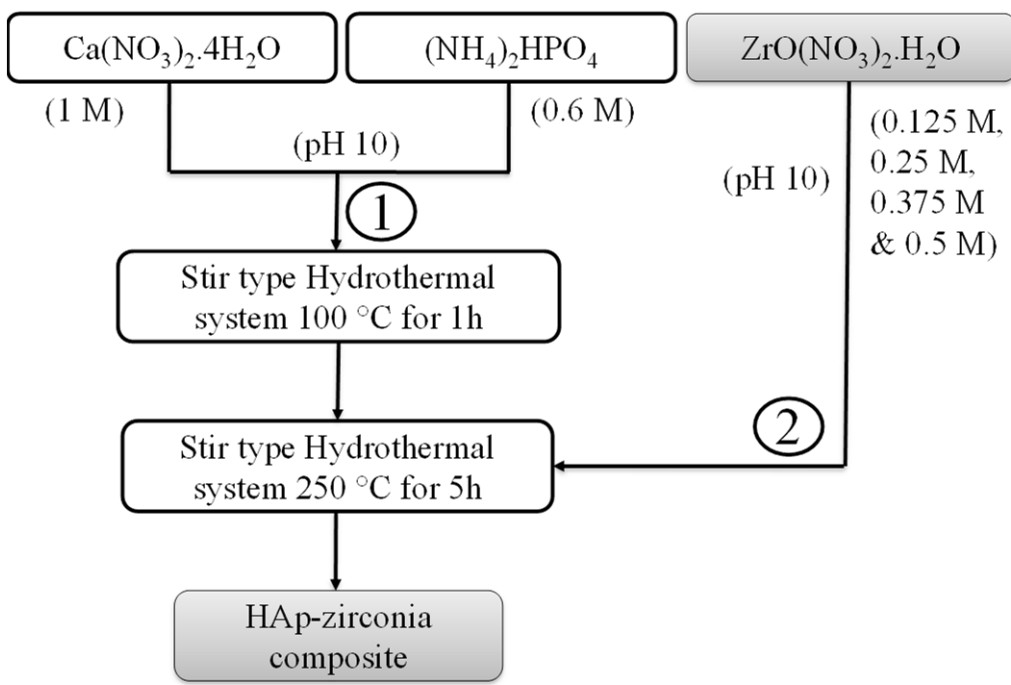

**Scheme 1.** Flowchart for the synthesis of the HAp-Zirconia composite.

Typically, 250 mL of 1 M $Ca(NO_3)_2 \cdot 4H_2O$ and 250 mL of a 0.6 M $(NH_4)_2HPO_4$ solution were mixed and transferred to a 1000 mL Teflon-lined stainless steel autoclave and hydrothermally treated at 100 °C for 1 h. This mixture was then treated for 5 h at 250 °C with 100 mL of various molarities of $ZrO(NO_3)_2 \cdot H_2O$. The powder was then dried and calcined for 2 h at 850 °C. A similar process was followed at a volume of 100 mL for different concentrations of $ZrO(NO_3)_2 \cdot H_2O$ (0.125 M (HZ1), 0.25 M (HZ2), 0.375 M (HZ3), and 0.5 M (HZ4)) at constant starting material concentrations of HAp. The same experiment

conditions were used to prepare pure HAp or pure zirconia, with the exception of the addition of their composite precursors in the autoclave setup, with only the addition of $Ca(NO_3)_2 \cdot 4H_2O$ with $(NH_4)_2HPO_4$ and $ZrO(NO_3)_2 \cdot H_2O$ used with pure HAp and pure zirconia, respectively. The stoichiometric equation was used to determine the concentrations of HAp and zirconia in the final samples. As per the stoichiometric calculations, the reaction between $Ca(NO_3)_2 \cdot 4H_2O$ and $(NH_4)_2HPO_4$ yielded a product $(Ca_{10}(PO_4)_6(OH)_2)$ of 29.32 g (31.86% yield). In addition, $ZrO(NO_3)_2 \cdot H_2O$ yielded 49.43% of $ZrO_2$ as a product, wherein the presence of $ZrO_2$ in the HZ1, HZ2, HZ3, and HZ4 composite samples was 1.54, 3.08, 4.62, and 6.16 g, respectively. Table 1 shows the presence of HAp and zirconia (weight percent) in the as-prepared samples.

**Table 1.** The presence of HAp and zirconia content in the as-prepared samples.

| Sample Code | HAp (Weight %) | Zirconia (Weight %) |
|---|---|---|
| Pure HAp | 100 | 0 |
| Pure zirconia | 0 | 100 |
| HZ1 | 95.01 | 4.99 |
| HZ2 | 90.49 | 9.51 |
| HZ3 | 86.39 | 13.61 |
| HZ4 | 82.64 | 17.36 |

### 2.2. Characterization of the HAp-Zirconia Nanocomposite

The obtained powder was characterized using an X-ray diffraction (XRD) Xpert powder diffractometer (PAN Analytical, Eindhoven, The Netherlands) with Cu-kα radiation of 1.5406 Å at a scan rate of 5 °/min in the scan range of 20–80° to determine the characteristic phase formation of the HAp-Zirconia composite. Thermal analyses such as thermogravimetry (TG) and differential scanning calorimetry (DSC) analyses of the prepared samples were performed from room temperature to 1200 °C at a heating rate of 10 °C/min using the NETZSCH STA 449F3 instrument (Yokohama, Japan). The morphological distribution of the prepared composite was determined using a transmission electron microscope (TEM, JEOL JEM 2100, Freising, Germany) and a field emission scanning electron microscope (FESEM, SIGMA HV (Carl Zeiss, Jena, Germany) with a Bruker Quantax 200–Z10 EDS detector, (Billerica, MA, USA)).

Green pellets of the HAp-Zirconia (HZ1) composite with dimensions of 12 mm in diameter and 2 mm in thickness were prepared using a hydraulic pellet press with an applied load of 12 MPa. No binder was added in the preparation of the composite pellet. The hardness and Young's modulus of the prepared composite were determined using an Ubi 1 Scanning Quasistatic Nanoindenter (TI-700; Hysitron Inc., Billerica, MA, USA) with a Berkovich pyramidal indenter. The surface area, pore size, and pore size distribution were measured using a BELSORP Max (Microtrac BEL, Osaka, Japan) utilizing the Bruner–Emmet–Teller (BET) and Barrett–Joyner–Halenda (BJH) techniques, and the experiments were performed under liquid nitrogen ($-196$ °C). The total pore volume was derived from the quantity of nitrogen adsorbed at $P/P_0 \sim 0.99$, while the surface area was estimated at a relative pressure ranging from 0.1 to 0.5 using the BET method. Before examination of the surface area, the samples were warmed at 250 °C for 2 h under a vacuum to eliminate the interlayer humidity and the volatile compounds of the samples.

The sintering process was carried out in an atmosphere of air. The samples were placed in an alumina crucible and transferred to a muffle furnace. The heating rate for the sintering process was maintained at 5 °/min and the system was allowed to cool down to room temperature automatically. The MG-63 human osteosarcoma cell lines were obtained from the National Centre for Cell Science (NCCS), Pune, India, and used in the cytotoxicity investigation utilizing the MTT assay. The cells were cultured in Eagle's minimal essential medium with 10% fetal bovine serum (FBS). In a $CO_2$ incubator, the cells were kept at 37 °C, 5% $CO_2$, 95% air, and 100% relative humidity. The cultures were switched every week, while the medium was switched twice a week. The cells were treated with different

concentrations (12.5, 25, 50, 100, and 200 µg) of the composite powder for 48 h. Cells treated without nanoparticles were used as a control. The percentage of cell viability was analyzed by the MTT assay as mentioned in our previous report [30].

## 3. Results

### 3.1. X-ray Diffraction Analysis

The XRD patterns of HAp, zirconia, and the HAp-Zirconia composites are shown in Figure 1. The characteristic peaks of HAp and zirconia were compared with standard JCPDS files. A hexagonally structured HAp with a primitive lattice associated with JCPDS file No. 09-0432 and an orthorhombic-structured zirconia with a primitive lattice correlated with JCPDS file No. 87-2105 were obtained [31,32]. With the aim of obtaining distinct phases of HAp and zirconia, the composite powder was calcined at 850 °C for 2 h [21]. The prepared HZ1 sample shows distinct peaks of both HAp and zirconia, which indicates the formation of a HAp-Zirconia composite compared with the other composite samples (HZ2–HZ4). Conversely, in HZ2–HZ4, the increasing concentration of zirconyl nitrate showed an increased trend in intensity (Figure 1b), whereas the reduction in the peak intensity of HAp could be attributed to alteration of the texture of HAp with respect to the concentration of $ZrO_2$ [33]. The particle size of the prepared samples was determined using the Scherrer equation. The average particle size for pure HAp, pure zirconia, HZ1, HZ2, HZ3, and HZ4 was 17.31 nm, 14.37 nm, 13.94 nm, 13.97 nm, 14.21 nm, and 14.16 nm, respectively.

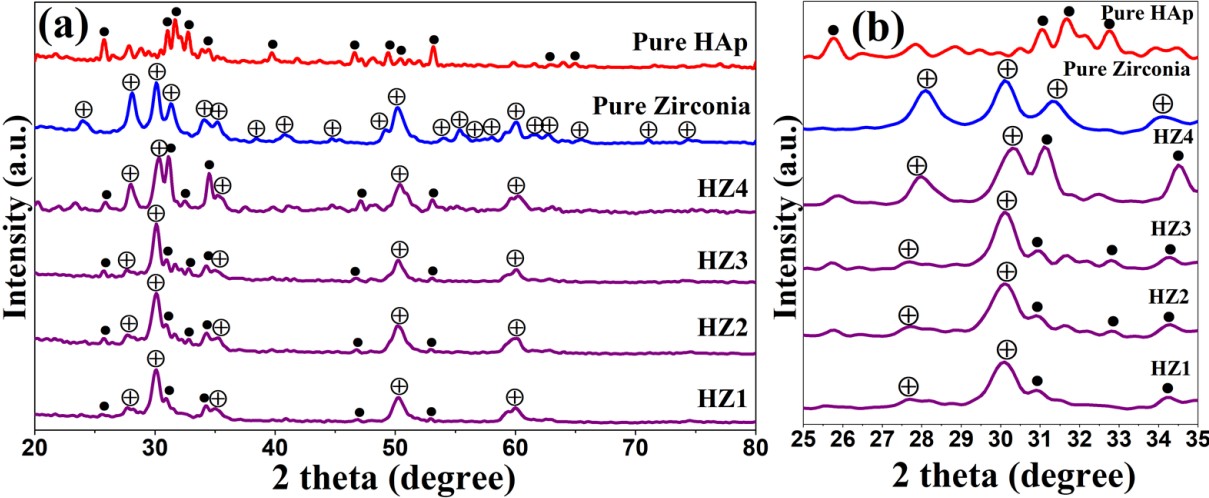

**Figure 1.** (**a**) XRD patterns of HAp, zirconia, and the HAp-Zirconia (HZ1–HZ4) composites (**b**) XRD patterns in the region of the major intensity peaks.

### 3.2. Electron Microscopy Analysis

The morphology of the prepared pure HAp and HZ1 composite powder was analyzed using TEM and FESEM (Figure 2). The TEM images illustrate the distribution of nanoparticles with a nonuniform morphology for pure HAp (Figure 2a). Figure 2c shows the FESEM image of the as-prepared HAp-Zirconia composite (HZ1); the formation of bulk agglomerates can be observed. The image at higher magnification revealed the presence of minute freely dispersed particles attached to the sample's surface. Figure 2b,e shows the HRTEM images of the pure HAp and the HAp-Zirconia composite (HZ1), respectively. The obtained d-spacing value of 0.28 nm corresponds to the (211) hkl plane of pure HAp, and 0.241 nm is associated with the (111) hkl plane of zirconia present in the composite, which are in good agreement with the XRD results.

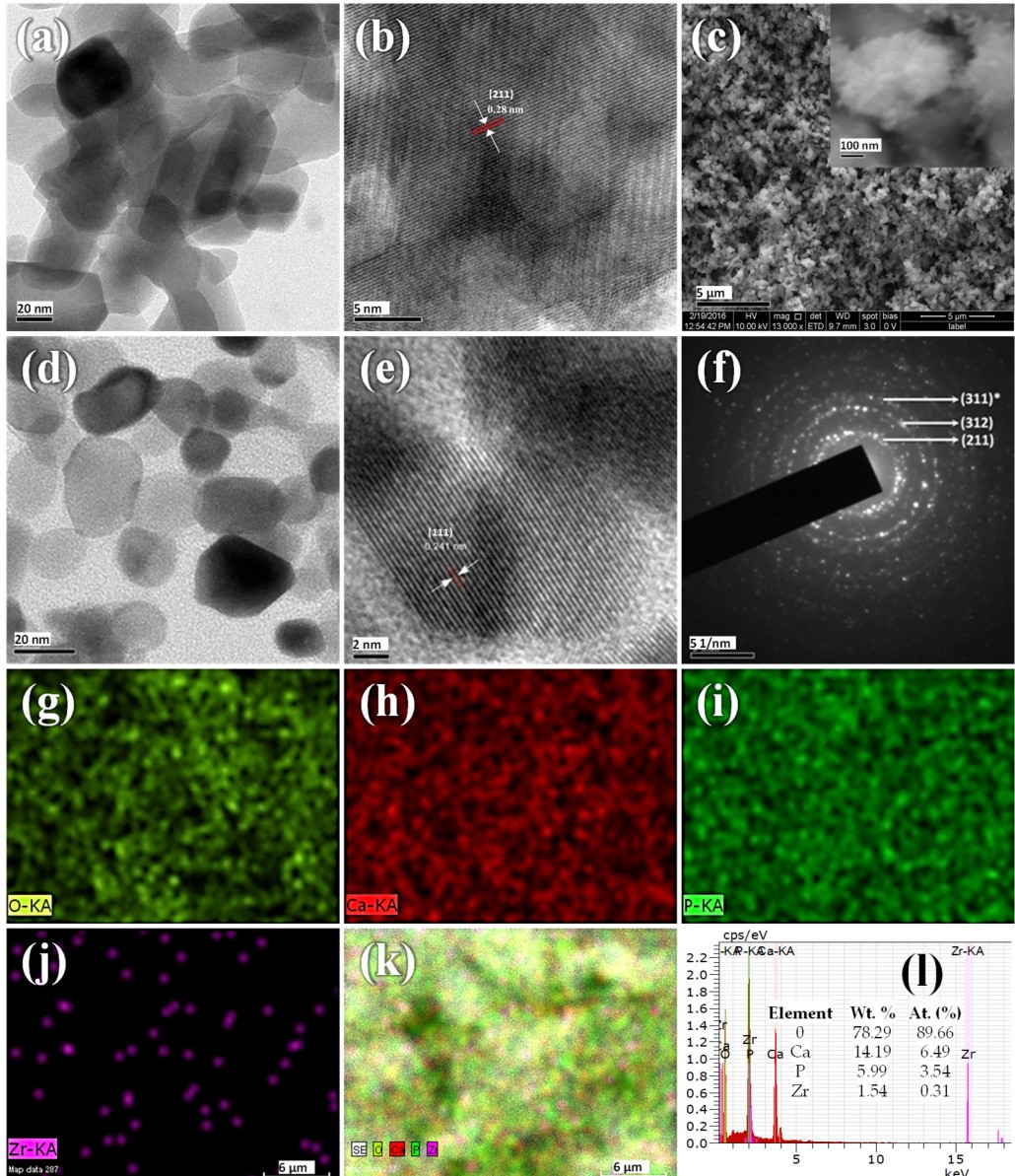

**Figure 2.** (**a**) TEM and (**b**) HRTEM images of pure HAp; (**c**) FESEM (**d**) TEM (**e**) HRTEM, and (**f**) SAED patterns of the HAp-Zirconia (HZ1) composite; (**g**–**k**) elemental mapping; and (**l**) energy dispersive spectrum with the percentage of chemical elements present in the HAp-Zirconia (HZ1) composite.

The elemental distribution of the prepared composite (HZ1) powder was determined using elemental mapping, as shown in Figure 2g–k. This confirmed the major distribution of Ca, P, and O in the middle region, which corresponded to HAp, and the sparse distribution of Zr in between Ca, P, and O. This illustrated the formation of a HAp-zirconia composite with a major HAp component that acts as a matrix and trace zirconia as a reinforcement material. The EDS spectrum (Figure 2l) revealed a <10% addition of zirconia to the HAp matrix. The atomic percentages of Ca, P, Zr, and O were 6.49, 3.54, 0.31, and 89.66%, respectively.

### 3.3. TG/DSC Analysis

The thermal behavior of the prepared HAp-Zirconia composite material was determined using the TG/DSC curve shown in Figure 3. In the temperature range of 30–100 °C, a 10% proportional weight loss was observed. Further, in the second stage, a weight loss of 15%, and in the third stage, a weight loss of 10% was observed in the temperature range of

100–250 °C and 250–450 °C, respectively. The DSC results revealed endothermic peaks at temperatures of 103 °C and 517 °C, and also an exothermic peak at 290 °C.

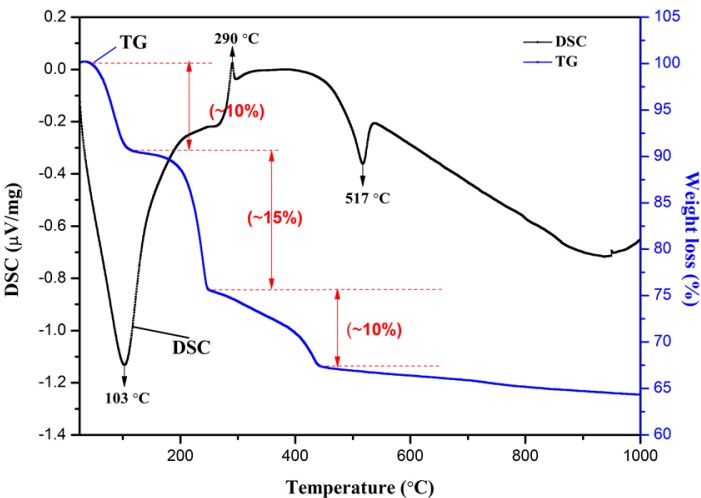

**Figure 3.** TG/DSC curve of the HAp-zirconia (HZ1) composite.

### 3.4. BET Analysis

Figure 4 shows the $N_2$ adsorption–desorption plot of the as-prepared composite sample. According to IUPAC, the obtained $N_2$ adsorption–desorption curves are associated with a Type II isotherm, which corresponds to the nonporous nature of the materials [34,35]. The resulting composite material's modest surface area of 19 $m^2$/g further validates this conclusion. The pore volume and pore diameter were calculated as 0.030152 $cm^3$/g and 14.27 nm, respectively.

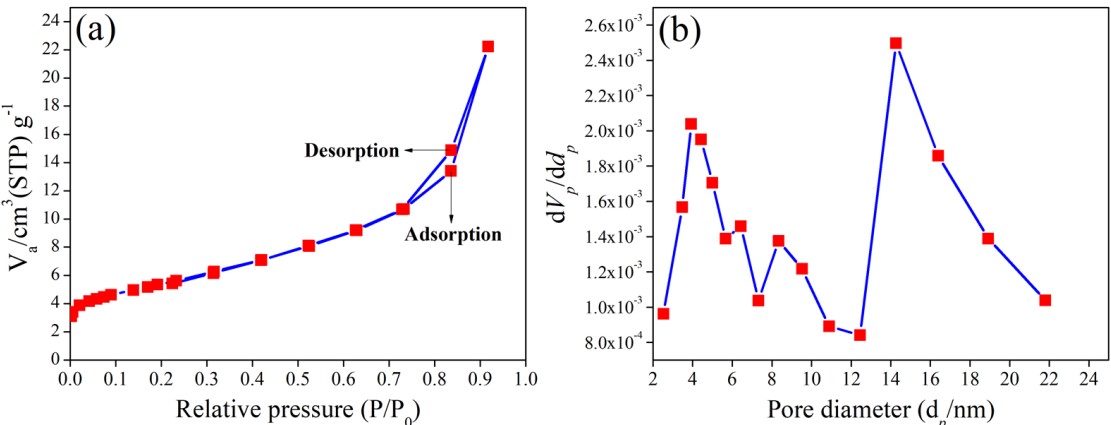

**Figure 4.** (**a**) $N_2$ adsorption–desorption isotherm and (**b**) BJH plot of the HAp-zirconia (HZ1) composite.

### 3.5. Nanoindentation Study

The mechanical behavior of the prepared composite was determined using a nanoindentation study. The load vs. displacement graph obtained from the nanoindentation study is shown in Figure 5. The hardness and Young's modulus of the unsintered pellet were calculated as 0.3 GPa and 2.26 GPa, respectively. The hardness of pure HAp was found to be 0.8 GPa. The hardness and Young's modulus values were found to be 0.9 GPa and 8.19 GPa, respectively [36].

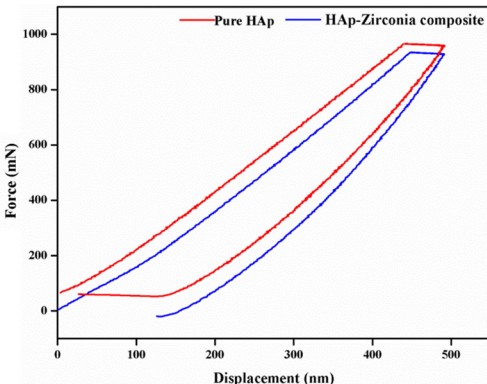

**Figure 5.** The force-displacement curve of pure HAp and the HAp-Zirconia (HZ1) composite.

### 3.6. In Vitro Biocompatibility

An MTT assay was performed to analyze the biocompatibility of pure HAp and the HAp-Zirconia (HZ1) composite powder (Figure 6). Even at the increased concentration of the HAp-zirconia composite, only a modest decrease in MG-63 cell survival was seen. The graph between the percentage of viable cells and different concentrations of the composite powder shows that 80% cell viability for the maximum concentration (200 µg/mL) tested against MG-63 cells, as shown in Figure 6a. After 48 h, no significant morphological change in the treated MG-63 cells was observed in the optical micrographs, as shown in Figure 6b.

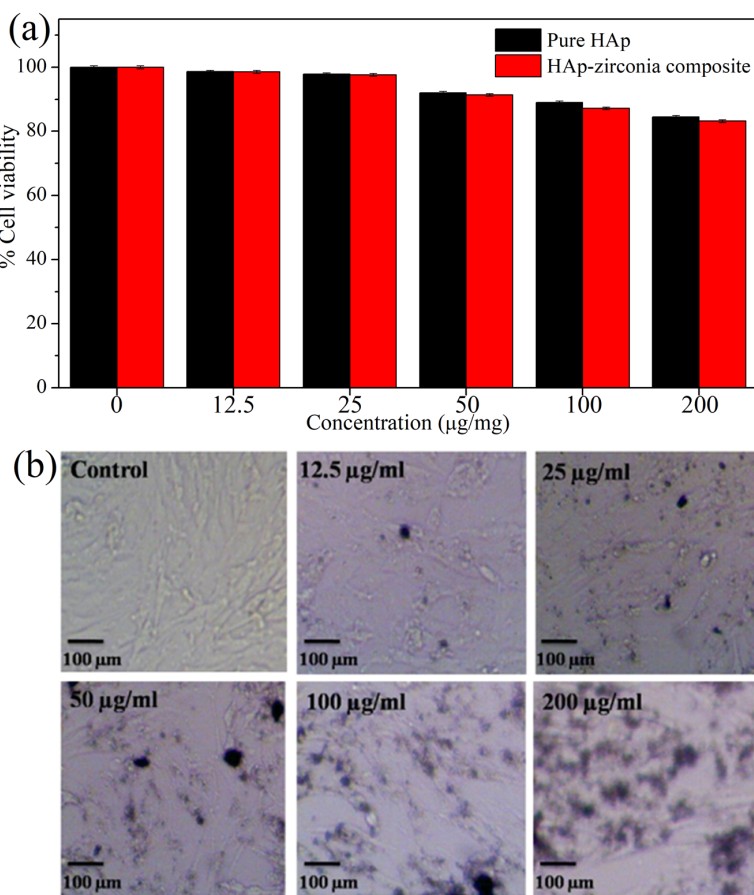

**Figure 6.** (**a**) MTT assay of pure HAp and the HAp-Zirconia (HZ1) composite, and (**b**) optical micrographs of the MG-63 cell line treated for 48 h with different concentrations of the HAp-Zirconia (HZ1) composite.

## 4. Discussion

From the XRD patterns (Figure 1), it was observed that at a higher concentration of $ZrO_2$, the peaks associated with zirconia also increased by around 28–32° (2θ value), which corresponded to the overlapping of the HAp-Zirconia peaks. Furthermore, no intermediate peaks were observed, which confirmed that either HAp does not decompose or that no interfacial reactions occurred between HAp and zirconia. Therefore, it was decided that the composition used for preparing the HAp-Zirconia composite (HZ1 sample) under hydrothermal conditions was the optimized condition. Therefore, only HZ1 was used further for the morphological, thermal, mechanical, and cytotoxicity analyses.

The formation of bulk agglomerates was observed in the FESEM analysis due to the composite's primary nucleation (Figure 2c) [37]. However, under higher magnification, the sample surface was observed to have an attachment of small-sized freely dispersed particles due to secondary nucleation. According to the TEM results in Figure 2d, the size of the HZ1 composite powder was observed to be in the 15–20 nm range, which confirmed the formation of nanocomposites. HAp has an oval and rounded hexagonal morphology, and zirconia has a spherical morphology. The SAED pattern shown in Figure 2f confirmed the corresponding planes of the HAp (211) and zirconia (311) phases, as discussed in the XRD analysis, which reaffirmed the formation of the HAp-Zirconia composites. A proper distribution of zirconia on the surface of HAp was observed in the elemental mapping results (Figure 2g–k). This showed that the majority of the HAp served as the matrix and that trace amounts of zirconia served as the reinforcement. Further, the EDS spectrum (Figure 2l) revealed the ideal percentage of zirconia (<10%) added to the HAp matrix which prevented Hap's decomposition at higher temperatures [38].

The $N_2$ absorption–desorption results (Figure 4) revealed the Type II isotherm that causes capillary condensation. The surface area, pore volume, and pore diameter were calculated as 19.319 $m^2/g$, 0.030152 $cm^3/g$, and 14.27 nm, respectively. This indicates that the multilayer adsorption of the prepared composites limits the high uptake of $P/P_0$ [39]. However, the moderate surface area of the composite powders might help with molecular attraction for biological applications [40].

According to the TG/DSC results (Figure 3), a relative weight loss of 10% at around 30–100 °C was observed, which corresponded to the vaporization of the adsorbed water [27]. The second stage of weight loss of up to 15% was attributed to the removal of hydroxyl groups adsorbing on the surface of the zirconia [41]. The third stage of weight loss (up to 10%) corresponded to the release of chemically bound water and the breakdown of the organic matrix and carbonates; moreover, the phase transition of zirconia from the amorphous to the tetragonal phase occurs in this temperature region [42–44]. When zirconia was added, the exothermic peak associated with the thermal conductivity of HAp, which typically appeared at about 400 °C, was diminished [21]. The endothermic peak observed at 103 °C corresponded to moisture evaporation, while the exothermic peak observed at 290 °C was associated with the decomposition and degradation of organic residues. Additionally, the endothermic peak associated with the crystallization of HAp was obtained at a temperature of 517 °C. The sharpness of the endothermic peak indicated a high degree of crystallinity [45]. Furthermore, the results confirmed a respectable thermal stability for the as-prepared HAp-Zirconia composite.

From the nanoindentation study (Figure 5), for the HAp sample, the obtained hardness was 0.8 GPa, which increased to 0.9 GPa with a Young's modulus of 8.19 GPa in the HAp-Zirconia composite sample. The results illustrated that the addition of zirconia increased the mechanical strength of the HAp matrix [30]. Moreover, the results indicated that the improved mechanical stability could help in mechanically bearable environments.

The MTT assay results revealed a good cell viability of 80%, even at a high concentration of 200 μg/mL of the sample. However, a marginal decrease in the biological activity was obtained for the composite materials compared with the pure HAp, especially at higher concentrations of the sample. This might be due to the physical impact of the composite material on the cell lines in powder form. Additionally, the higher concentration increases the

material's physical stress in its composite state, which reduces the biological activity of the HAp/zirconia composite. However, 48 h of testing on MG-63 cells (Figure 6b) revealed no significant morphological changes, confirming the biocompatibility and demonstrating the potential of the HAp-Zirconia composite as a potential material for bone-related applications.

## 5. Conclusions

A HAp-Zirconia composite was successfully prepared using a specially designed hydrothermal reactor under optimized conditions. Furthermore, the prepared materials were systematically studied for their physicochemical, thermal, and mechanical properties. Moreover, a biocompatibility evaluation was also carried out carefully. The XRD patterns revealed that HAp peaks were dominant at a lower zirconyl nitrate concentration (0.125 M), whereas zirconia peaks were dominant at a higher concentration (0.5 M). This clearly indicated that the increase in the molar concentration of zirconyl nitrate altered the reinforcement material (zirconia) in the matrix (HAp) in the prepared composites. Further, the elemental mapping of the composite prepared at a 0.125 M concentration of zirconyl nitrate confirmed the reinforcement of zirconia nanoparticles in the HAp matrix. The mechanical study indicated the high-temperature (1200 °C) processing stability of the developed composite material, which could be used in high-strength bearing circumstances. Additionally, the MTT assay showed 82% cell viability against a high concentration (200 μg/mL) of the HZ1 composite material, revealing biocompatibility that could be applicable for bone implant applications.

**Author Contributions:** Conceptualization, V.R.S. and R.M.; methodology, V.R.S. and K.A.; software, V.R.S. and K.A.; validation, V.R.S. and K.A.; formal analysis, V.R.S. and K.A.; investigation, V.R.S. and K.A.; resources, V.R.S. and R.M.; data curation, V.R.S. and K.A.; writing—original draft preparation, V.R.S. and K.A.; writing—review and editing, R.M., V.P. and T.O.; visualization, R.M., V.P. and T.O.; supervision, R.M. and T.O.; project administration, R.M., T.O.; funding acquisition, R.M., V.P. and T.O. All authors have read and agreed to the published version of the manuscript.

**Funding:** This research was funded by National Research Foundation of Korea (NRF) grant funded by the Korean government (MSIT), grant number No. 2022R1A2C1004283.

**Institutional Review Board Statement:** Not applicable.

**Informed Consent Statement:** Not applicable.

**Data Availability Statement:** Not applicable.

**Conflicts of Interest:** The authors declare no conflict of interest.

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
