# Peer review of "Synthesis of Hydroxyapatite (HAp)-Zirconia Nanocomposite Powder and Evaluation of Its Biocompatibility: An In Vitro Study"

_applsci, doi:10.3390/app122111056_

Round 1

Reviewer 1 Report

1)    The authors claimed, “Nevertheless, their limited ability to withstand thermal processing and weak mechanical strength prevents them from being used in hard tissue engineering.”. Nevertheless, the HAP or CaP possesses good thermal processing capability and biomechanical properties. The authors should raise examples, references, or trials to demonstrate the drawback of HAP or CaP described in the manuscript. 

2)    The Introduction should be divided into part sections to highlight the importance the authors would like to delineate.

3)    Furthermore, from the Introduction, the reviewers could not find what problems, hypotheses, or unmet medical needs the authors would like to address.

4)    The Materials and Methods should be clearly described, such as how many molars or quantity or concentration or examining protocol or vacuum the authors used in the current study.

5)    The Results should be solely described in one part, separate from the Discussion, and delineate the experimental results.

6)    The Discussion should be solely placed in one section rather than combined with the Results. Moreover, importantly, the reviewers were disappointed with the Discussion. The reviewers could not find any improvement, scientific advancement, or novelty in the current study. Therefore, the authors must delineate the comparative results with previous arts or publications and demonstrate the current study’s superiority, rather than only phenotypic observation and no interpretation of findings in the current study. 

Reviewer 3 Report

See the attached file for the comments.

Round 2

Reviewer 1 Report

The authors replied to almost all the comments the reviewers raised. However, the authors still did not face the unmet scientific problems: "The reviewers could not find any improvement, scientific advancement, or novelty in the current study. Therefore, the authors must delineate the comparative results with previous arts or publications and demonstrate the current study's superiority, rather than only phenotypic observation and no interpretation of findings in the current study." Furthermore, the authors delineated an alternative time-saving approach for synthesizing zirconia-HAp, which seems not a complex modification and a critical step for this product to be used in industrial applications in the future. But, I think it is acceptable for publication in the Applied Sciences. Finally, please carefully check the format and smoothness and fluency, especially the Discussion.
